# Evolutionary origins and life-history correlates of coloniality in the epiphytic fern genus *Platycerium* (Polypodiaceae)

**Riccardo Ciarle** ORCID*, **Katrijn de Bock** ORCID, **Kevin C. Burns**

Te Kura Mātauranga Koiora, School of Biological Sciences, Te Herenga Waka, Victoria University of Wellington, Wellington, Aotearoa, New Zealand

* riccardo.ciarle@vuw.ac.nz

## Abstract

Many animals live in cooperative groups comprised of morphologically differentiated individuals that subdivide labour to help the group persist in harsh, unpredictable environments. Recently it has been shown that a colonial fern from Australasia, *Platycerium bifurcatum* (Polypodiaceae), sub-divides labour similarly, with individuals producing morphologically different fronds depending on their vertical position within the colony. The genus contains approximately 18 taxa, which range from solitary to colonial. Whether other *Platycerium* species exhibit similar morphological differentiation remains poorly understood, and the evolutionary origins of coloniality along with its life-history correlates across the genus remain unknown. Here, we use ancestral state reconstruction to explore the evolution of coloniality and morphologically differentiated division of labour in the genus *Platycerium*. We found coloniality to be likely ancestral in *Platycerium,* with the condition being lost twice across the phylogeny. Eight *Platycerium* species exhibited colonies with morphologically differentiated individuals. This condition is derived and likely evolved twice within the genus. Coloniality was also negatively correlated with nest frond length and width but was unrelated to strap frond length. Overall, results reveal the evolutionary origins and life-history correlates of coloniality across the genus *Platycerium*, and support a scenario in which a colonial species with morphologically variable colony members evolved gradually from a solitary species.

## Introduction

Many animals form cooperative groups to enhance individual survival in harsh, unpredictable environments. Individuals within the group can sometimes be morphologically differentiated and specialise on performing different tasks. This condition is considered one of the most complex forms of social behaviour [1–3], and has evolved independently in several groups of animals, including Hymenoptera [2,4,5], Blattodea

**Data availability statement:** All relevant data are within the paper and its Supporting Information files.

**Funding:** This study was funded by the Royal Society of New Zealand Marsden Fund (E3888, Eusociality in plants). The funders had no role in study design, data collection and analysis, decision to publish, or preparation of the manuscript.

**Competing interests:** The authors have declared that no competing interests exist.

[6,7], Crustaceans [8,9], Mammals [10,11], and others [12,13]. But while widely known in animals, it is relatively unknown in plants (but see [14]).

Recently it has been shown that an epiphytic fern (i.e., a fern that grows on trees) from Australasia, *Platycerium bifurcatum* (Polypodiaceae), also forms cooperative colonies within which individuals subdivide labour. The species shows within-individual frond dimorphism, meaning that each individual produces sterile "nest" fronds that senesce almost immediately after maturation but do not abscise after death [15], and reproductive "strap" fronds which are typically long, narrow, and lobed, and remain photosynthetic at maturity [14]. Each colony also contains morphologically differentiated individuals (termed morphologically variable colony members hereafter) (Fig 1). Individuals at the bottom of colonies produce small nest fronds and pendulous strap fronds, and individuals at the top produce larger, more lobed nest fronds and upright, stiffer strap fronds [16,17].

These morphological differences have been interpreted as evidence of cooperative division of labour [16,18,19]. Specifically, the larger nest fronds and upright, stiffer strap fronds of higher individuals facilitate water and debris harvesting, while the smaller nest fronds of lower individuals allow for storage, protection and grip onto the host. The other 17 *Platycerium* species that are currently recognised are all within-individual frond dimorphic [20]. Six species are solitary, but 11 have been previously described as forming colonies similarly to *P. bifurcatum* [20] (Fig 2). Despite this, whether any of them exhibits morphologically variable colony members remains unknown.

*Platycerium* is a popular ornamental that is currently distributed in subtropical to tropical regions, with 11 species endemic to southeast Asia and Australasia, 6

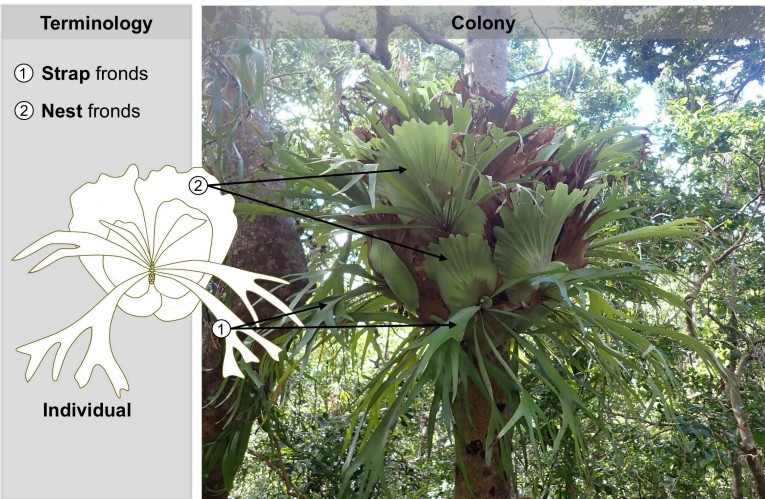

**Fig 1. Example of a colony of *Platycerium bifurcatum*.** Each individual produces two types of fronds: sterile nest fronds that senesce almost immediately after maturation but do not abscise after death, and reproductive strap fronds which are typically long, narrow, and lobed, and remain photosynthetic at maturity. Different individuals tend to produce morphologically variable nest and strap fronds dependening on the position along the vertical grandient of the colony.

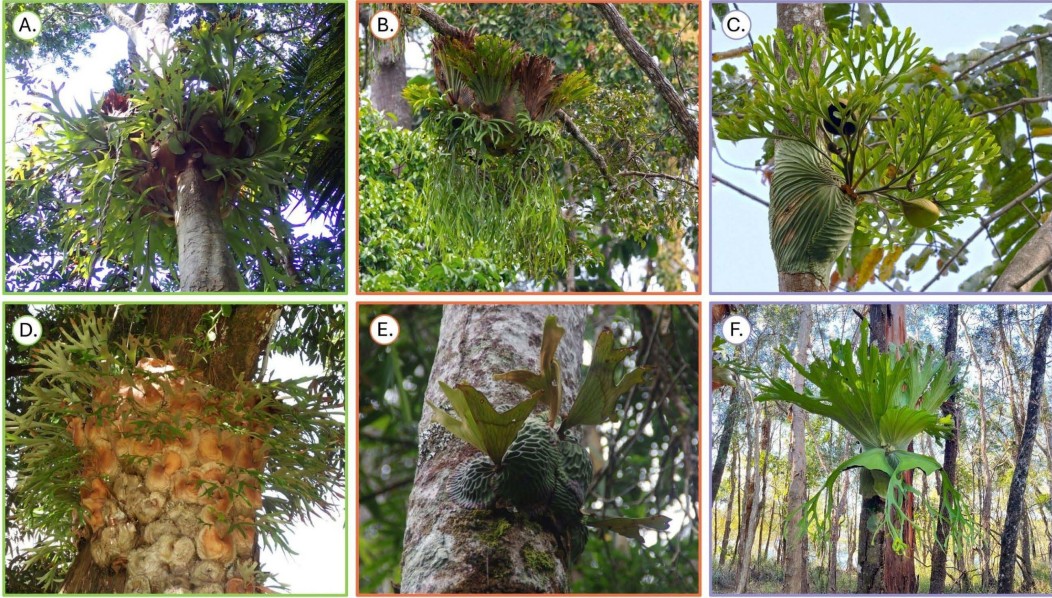

**Fig 2. Examples of colonial and solitary *Platycerium* species.** The genus currently comprises 18 species distributed across subtropical and tropical regions, from Peru to Madagascar to Indochina and Australia. All species are within-individual frond dimorphic, and produce sterile "nest" fronds and reproductive "strap" fronds. **A)** *P. bifurcatum*; **B)** *P. coronarium* © ayuwat; **C)** *P. ridleyi* © ayuwat; **D)** *P. alcicorne* © Jean-Philippe; **E)** *P. madagascariense* © Vincent Porcher; **F)** *P. superbum* © Trina. All photos, except for *P. bifurcatum* were taken from iNaturalist.

species endemic to tropical Africa and Madagascar, and one species endemic to Bolivia and Peru [21–23]. This wide ecological and geographic range of the genus facilitates the study of transitions into and out of coloniality, the evolution of morphologically variable colony members, and their life-history correlates. Yet, the evolutionary origins of coloniality and morphologically variable colony members in *Platycerium* remain poorly understood, and whether transitions in and out of coloniality are associated with other life-history traits is unknown.

Several traits may correlate with coloniality in *Platycerium*. I) Solitary species may produce larger strap and nest fronds to improve litterfall entrapment, as they cannot count on a "communal" nest [16]; II) In individuals of colonial species, sterile and fertile segments of strap fronds may develop simultaneously, as these individuals can count on shared resources and they would benefit from early reproduction. III) Previous work suggests that in solitary species nest fronds may not senesce at maturity and remain green and photosynthetic [20]; IV) Individuals of colonial species may lack water storage tissues, which would become redundant if the communal nest in general retains enough moisture and nutrients.

In this study, we first determine which *Platycerium* species possess morphologically variable colony members. Using ancestral state reconstruction we then explore the evolutionary history of within-individual frond dimorphism, coloniality and morphologically variable colony members across the *Platycerium* phylogeny. Finally, we use mixed effect models in a Bayesian framework to determine the life-history correlates of coloniality across the *Platycerium* phylogeny.

## Methods

### Definition and characterization of coloniality and morphologically variable individuals

First, we defined each *Platycerium* species as either solitary or colonial based on Hoshizaki and Price (1990). Colonial species can proliferate from the roots, producing "pups" which then become independent. The only exception is *P. coronarium,* which was categorised as colonial but produces offsets from rhizome branches, not from the roots [20].

A species was defined as within-individual frond dimorphic if it was either solitary or colonial with each individual producing distinct fertile and sterile fronds (i.e., strap and nest fronds). A species was classified as colonial with morphologically variable colony members if individuals within a colony were morphologically differentiated. Specifically, each colonial species was categorised as being either monomorphic (M), strap frond variable (SV), nest frond variable (NV) or both strap and nest frond variable (SNV). A species was classified as 'M' if, within a colony, individuals were not morphologically differentiated along the colony's vertical gradient. A species was categorised as 'SV' if, within a colony, strap fronds were morphologically differentiated among individual colony members. More specifically, when a significant positive association between height within colony and strap frond angle was observed. As mentioned above, this indicates that higher individuals are better adapted to rainwater harvesting. A species was categorised as 'NV' if, within a colony, an individuals' nest fronds were morphologically differentiated, showing a significant positive association between height within colony and nest frond area. Similarly to strap fronds, this indicates that higher individuals are better adapted to rainwater harvesting while lower individuals specialize in water storage and attachment to the host. Finally, a species was categorised as 'SNV' if it was both SV and NV. This method was defined based on previous observations of *P. bifurcatum* [16,18].

To classify species into these categories, data for strap frond angle and nest frond area were gathered from photos. For all species, we downloaded all available photos from the gbif and iNaturalist (iNat) repositories using the R package *rgbif* [24]. We then established four criteria for inclusion: 1) The photo had to be taken in the wild of a specimen not growing on artificial support, within the native range of the species. 2) The camera and the specimen had to be at comparable heights, so that the colony was visible lengthwise in its entirety. 3) At least one nest frond from the bottom, the middle and the top of the colony had to be clearly visible. 4) At least one strap frond from the bottom, the middle and the top of the colony had to be clearly visible lengthwise and positioned perpendicularly to the camera. After selection, all photos were manually screened to assess taxonomic accuracy. Criteria 1 and 2 were required to correctly assess the presence of morphologically variable colony members, and photos were discarded if they did not meet them. Conversely, if criterion 3 was not met, the photo was used to assess SV. Similarly, if criterion 4 was not met, the photo was used to assess NV.

Nest frond area was calculated by delineating the perimeter of each frond and then measuring the enclosed area. Strap frond angle was measured by assigning a value along the range of 180° to vertical upright fronds, 90° to horizontal fronds and 0° to vertical pendulous fronds (Supplementary 1, Figures S1-2 in S1 File). Frond height was measured as the distance of each frond from the lowest point of the colony. All measurements were conducted on ImageJ [25].

Finally, we assessed the presence of morphologically variable colony members based on nest and strap fronds. We tested for significant associations between height and frond morphology by using Bayesian generalised mixed effect models in the R package *brms* [26]. We used strap frond angle and nest frond area as dependent variable and height within colony as predictor variable. *Brms* uses the probabilistic programming language Stan, allowing to account for multi-level data (e.g., fronds belonging to multiple colonies), small sample sizes and non-normally distributed data [26]. Each model was run using 4 chains, each with 20000 iterations and 1000 warmup. Results were validated using the Rhat, Bulk ESS and Tail ESS parameters, which assess the quality and reliability of the posterior distribution in Bayesian models.

## Phylogenetic analysis and ancestral state reconstruction

For the *Platycerium* phylogeny, we built a maximum likelihood tree using plastome data sequenced by Xue et al. (2024) including all 18 species of *Platycerium* plus *Hovenkampia schimperiana*, six *Pyrrosia* species and *Thylacopteris papillosa* as the outgroup. Sequences were aligned using iterative refining methods in MAFFT [27], and no trimming was performed. The best model of evolution was selected based on the corrected Akaike information criterion (AICc) using JmodelTest version 2.1.10 [28]. Maximum likelihood analyses were performed using IQ-TREE version 2.3.4 [29]. Branch support was found using 10000 iterations of ultrafast bootstrap approximation [30].

To determine the evolutionary history of within-individual frond dimorphism and colonies with morphologically variable colony members across *Platycerium*, we used marginal ancestral state reconstruction for discrete characters. We ran two

separate analyses. For frond dimorphism we fitted a continuous-time Markov model of discrete character evolution using the R package *corHMM* [31]. For a two-state character (frond monomorphic/dimorphic), there are four possible models of evolution: equal-rates (ER) model, all rates different (ARD) model and two irreversible models, one allowing only 0–1 changes and the other allowing only 1–0 changes [32]. The model that best fit our data was selected based on AICc using the *fitMK* function in the *phytools* package [33]. To further validate the results, we used stochastic character mapping as an alternative approach [34], using the *make.simmap* function in the *phytools* package to simulating 1000 character histories with Bayesian MCMC under the best-fitting model [33].

For the ancestral reconstruction of coloniality and morphologically differentiated individuals, we defined 3 possible states: Solitary, Colonial with monomorphic colony members and Colonial with morphologically variable colony members. We then built a transition matrix and used the *fitMK* function to define a model of discrete character evolution (Supplementary 1, Table S3 in S1 File). Finally, we used stochastic character mapping with the *make.simmap* function to simulating 1000 character histories with Bayesian MCMC under the defined model [33].

### Morphological predictors of coloniality

To determine whether other morphological factors correlated with coloniality across the *Platycerium* phylogeny, we collated data for strap frond maximum length; nest frond maximum length and width. We also amassed data for categorical morphological traits, with each species classified based on whether sterile and fertile segments of fertile strap fronds develop simultaneously or in succession; the colour of mature nest fronds (either green and photosynthetic at maturity; brown and withering at maturity or both); the presence/absence of water storage tissue. Data for all morphological traits was gathered from Hoshizaki and Price (1990).

We then used a mixed-effects logistic regression model in a Bayesian framework with the R package *MCMCglmm* [35]. We ran 6 independent models. Coloniality (binary variable) was always used as the dependent variable, while each morphological trait was used as the sole predictor for its model. *MCMCglmm* uses a Markov Chain Monte Carlo (MCMC) method and allows to fit generalized linear mixed models with non-Gaussian response variables (e.g., dichotomous variables). It also allows to implement phylogenetic information into the model [36]. A non-informative prior was used for all analyses, running a single chain with 1010000 iterations and a burn-in period of 10000.

After model estimation, to assess the goodness of fit of the prior distribution used, we performed Prior Predictive Checks (PPC) in the R package *brms* [26]. The *brm* function was used to simulate data only from the prior distribution. The *pp_check* function was then used to visualize and compare the simulated data to the real data. We then used Kolmogorov-Smirnov test for significant differences between the prior-predictive simulated data and the actual observed data. P-values $> 0.05$ would indicate that the prior used for the *MCMCglmm* analysis aligns with the observed data [37]. Finally, for model selection we used the Deviance Information Criterion (DIC), which is used for comparing the fit of Bayesian models. Lower DIC values indicate better model fit relative to model complexity [38].

### Results

Brms models indicated that 8 species produced colonies with morphologically differentiated individuals. Specifically, *P. alcicorne*, *P. bifurcatum* and *P. willincki* are strap frond variable, and *P. stemaria, P. angolense, P. quadridichotomum, P. bifurcatum* and *P. willincki, P. veitchii* and *P. hillii* are nest frond variable. Only *P. bifurcatum* and *P. willincki* were found to be both nest and strap frond variable (Supplementary 1, Table S1 in S1 File). This is consistent with previous studies on *Platycerium bicurcatum*, which found individuals within colonies to produce morphologically distinct nest and strap fronds across the vertical gradient [16,18,39].

The maximum likelihood tree we generated was fully supported at each node (Supplementary 1, Figure S3 in S1 File), and generally consistent with previous studies [22], with the exceptions of *P. ellissi* being sister to *P. alcicorne* and *P. wallichii* being sister to *P. superbum* + *P. wandae*. Continuous ancestral state reconstruction showed that within-individual frond dimorphism

was most likely ancestral within *Platycerium* (Prob > 0.99) was never lost, and likely represents a synapomophy of the genus, as the trait is not present in its sister taxon *Hovenkampia*, and in most *Pyrrosia* species [40,41] (Fig 3). Similarly, the ancestor of all extant *Platycerium* species was estimated as probably colonial (Prob = 0.77, Table S2, Fig 4), possibly with morphologically undifferentiated individuals (Prob = 0.61, Table S2, Fig 4). Coloniality was also lost twice, once in *P. ridleyi* and once in the ancestor of *P. wallichii, P. superbum, P. wandae, P. grande* and *P. holttumii,* leading to 6 solitary species (Fig 4).

Coloniality with morphologically variable colony members was estimated as likely derived within *Platycerium* (P = 0.71, Table S2, Fig 4), and the condition evolved at least twice within the genus (Fig 4), once in the ancestor of *P. bifurcatum, P. willincki, P. veitchii* and *P. hillii,* and once in the ancestor of *P. quadridichotomum, P. madagascariense, P. alcicorne, P. ellisii, P. angolense, P. stemaria* and *P. andinum*. The condition was likely also lost twice, once in *P. madagascariense* and once in *P. andinum,* leading to 2 colonial but individual monomorphic species (Fig 4). Overall, results support a scenario in which colonial species with morphologically variable colony members evolved gradually from a colonial ancestor with monomorphic individuals, which itself evolved from a solitary, within-individual frond dimorphic taxon.

Finally, results of MCMCglmm models showed that, across the *Platycerium* phylogeny, coloniality negatively correlated with nest frond length and width but was unrelated to strap frond size (Table 1). This means that solitary *Platycerium* species tend to possess larger nest fronds but produce strap fronds like those of colonial species. Coloniality was also negatively associated to the successional development of sterile and fertile segments of fronds, meaning that in colonial species the two segments tend to develop simultaneously (Table 1). Priors used for all *MCMCglmm* models aligned with the observed data (Supplementary 1, Table S4 in S1 File).

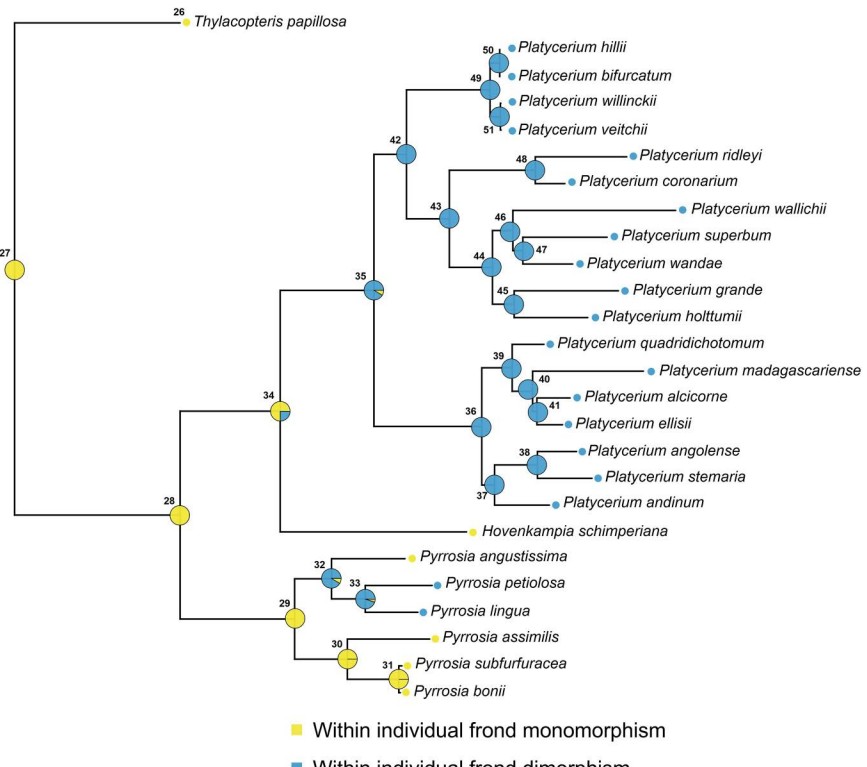

**Fig 3. Results of ancestral state reconstruction, showing the evolutionary history of within-individual frond dimorphism across the *Platycerium* phylogeny.** The ancestor of all extant *Platycerium* species was likely frond dimorphism, and frond dimorphism was never lost across the phylogeny.

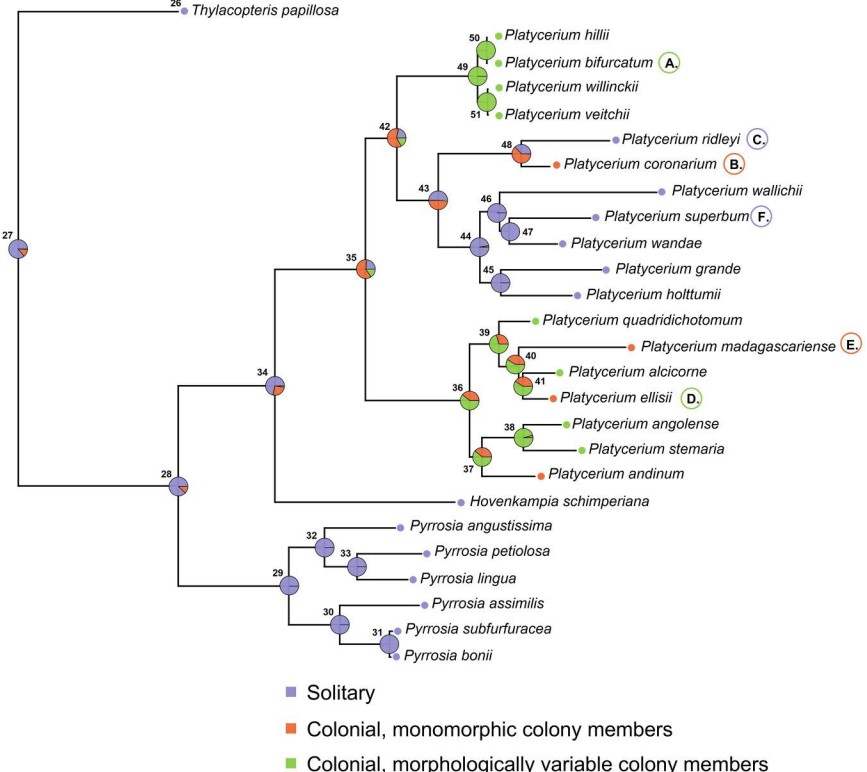

**Fig 4. Results of ancestral state reconstruction, showing the evolutionary history of coloniality across the *Platycerium* phylogeny.** The ancestor of all extant *Platycerium* species was likely colonial, with monomorphic colony members. The presence of morphologically variable colony members is a derived condition and evolved twice, once in the ancestor of *P. bifurcatum, P. willincki, P. veitchii* and *P. hillii,* and once in the ancestor of *P. quadridichotomum, P. madagascariense, P. alcicorne, P. ellisii, P. angolense, P. stemaria* and *P. andinum.* Coloniality was also lost twice, once in *P. ridleyi* and once in the ancestor of *P. wallichii, P. superbum, P. wandae, P. grande* and *P. holttumii.* Finally, coloniality with morphologically differentiated individuals was likely also lost twice, once in *P. madagascariense* and once in *P. andinum.*

**Table 1. Results of MCMCglmm models, always using coloniality as the dependent variable. Coloniality negatively correlates with nest frond length and width, and positively correlates with the simultaneous development of sterile and fertile segments of strap fronds. Coloniality is also unrelated to strap frond length, the colour of nest fronds at maturity and the presence of water storage.**

| DIC | Random Effect (species) | Residual Variance | Predictor | Estimate | 95% CI (Lower) | 95% CI (Upper) | Effective Sample | pMCMC |
|---|---|---|---|---|---|---|---|---|
| 0.821 | 167.2 | 67643 | Strap Length | −1.835 | −5.895 | 1.447 | 755.5 | 0.244 |
| 0.440 | 80.47 | 43366 | Nest Length | −5.216 | −11.031 | −0.430 | 1000 | 0.006 ** |
| 0.366 | 11,41 | 27455 | Nest Width | −8.778 | −18.017 | −0.639 | 1000 | 0.001** |
| 0.98 | 1677 | 65102 | Colour at maturity | 20.110 | −212.45 | 203.62 | 479.9 | 0.784 |
| 0.585 | 303.8 | 56511 | Development | −284.882 | −688.372 | 7.691 | 1000 | 0.030* |
| 1.579 | 1918 | 63903 | Water storage tissue | 99.56 | −362.05 | 631.27 | 1000 | 0.606 |

## Discussion

Within-individual frond dimorphism appeared to be ancestral in *Platycerium,* and the condition was never lost. This becomes clear when considering that in all *Platycerium* species, within-individual frond dimorphism goes well beyond reproduction, with strap and nest fronds performing considerably different functions. Solitary species produce either large,

fan-shaped nest fronds that collect water and flexible strap fronds that undergo photosynthesis and reproduction, or rounded nest fronds that clasp the branches of host trees for structural support and water storage, and stiff, erect, strap fronds that collect and channel rainwater down to nest fronds [18]. Colonial species seem to produce a combination of strap and nest frond morphology not seen in solitary species, but nest and strap fronds always retain distinct functions.

Coloniality was likely ancestral in *Platycerium*. Previous work in animals indicates that group living, cooperative behaviour and division of labour are often found in harsh and unpredictable environments [42,43], even in the absence of kin selection [44]. Epiphytes are typically exposed to harsh conditions at the tops of trees characterised by water limitation, nutrient deficiency, and strong irradiation [45]. Thus, harsh environmental conditions may have triggered the evolution of coloniality and cooperation in *Platycerium*.

Coloniality appears to have been lost in *Platycerium*, once in *P. ridleyi* and once in the ancestor of *P. wallichii, P. superbum, P. wandae, P. grande* and *P. holttumii.*, with all solitary *Platycerium* species confined to tropical Indochina [20]. The environmental conditions across their range are characterised by relatively stable temperatures and high levels of precipitation year round. It is possible that coloniality is no longer advantageous in such stable climates with low levels of environmental stress. Also, larger leaves are associated with warm, wet environments [46,47], allowing solitary *Platycerium* species to produce larger nest fronds (as our results indicate) to compensate for the absence of a communal nest.

We highlight that our results found that the ancestor of all *Platycerium* species could have been solitary with a 23% probability. Under this improbable but possible scenario coloniality would have appeared three times independently, while the solitary habit would have been maintained in the *P. superbum* group. Still, the causes behind the gain/loss of coloniality/solitary habit would remain consistent with those illustrated above.

Producing colonies with morphologically variable colony members is likely a derived trait within *Platycerium*. The condition is currently present in *P. hillii, P. bifurcatum, P. willicnkii, P. veitchii, P. quadridicotomum, P. alcicorne, P. angolense P. stemaria*, and likely evolved twice and was lost twice. As with coloniality, the evolution of morphologically differentiated colony members may be linked to harsh, unpredictable environments. But first, the species characterised as possessing morphologically variable individuals in this study need to be examined further to better quantify the individual morphological variation in nest frond size and strap frond angle, along with other morphological and physiological traits. So far this has only been (partially) done in *P. bifurcatum* [16]. Without this data any hypothesis about the ecological drivers of individual variability in *Platycerium* cannot be properly tested.

Solitary species possessed larger nest fronds. This could either indicate that larger nest fronds favour transitions to solitary living or the other way around. Regardless, solitary species may benefit from possessing larger fronds as this can improve litterfall entrapment and rainwater harvesting in the absence of a communal nest. This would explain why the increase in size is confined to nest fronds, as strap fronds perform functions (e.g., reproduction, photosynthesis) which are not apparently affected by the loss of coloniality. Solitary individuals also developed sterile and fertile segments of strap fronds in sequence. If all colonial species exhibited cooperative behaviour and division of labour [19], individuals within colonies could be able to invest in reproduction much sooner than solitary species, which first need to establish photosynthetic fronds and gather the necessary resources. Anecdotal evidence seems to confirm this, as *P. superbum* (solitary) has been observed producing many nest fronds before the first strap frond, while *P. bifurcatum* (colonial) was observed producing reproductive strap fronds from the early stages of development [20].

While individuals of colonial species possessed smaller nest fronds within *Platycerium*, we noted that all *Platycerium* species are relatively large when compared to their sister taxa (i.e., *Hovenkampia* and *Pyrrosia*). It is therefore possible that coloniality promotes larger size. It would be interesting to test this hypothesis not only in *Platycerium*, but in epiphytic and non-epiphytic ferns in general.

Overall, these results illustrate how coloniality and division of labour could have evolved within *Platycerium*. Research conducted so far on *Platycerium bifurcatum* illustrate how in this species coloniality and division of labour may arise from a relatively simple mechanism, in which multiple ramets/genets aggregate and develop differently depending on their

relative position within the colony [14,19,39]. Also, as *P. bifurcatum*'s rhizomes may develop downward (Burns, 2024; personal observation), the morphological variability of nest and strap fronds across the vertical gradient may result from a simple age difference between individuals at the top and bottom of colonies. We hope future work will extend the study of the origin of coloniality and the mechanisms behind it to other species of *Platycerium*, to shed light on how coloniality and division of labour can develop in plants.

In this study, whether a colonial species possessed morphologically variable colony members was determined from photos. This inevitably led to uncertainties in the results, and while we established several selection criteria to minimize this uncertainty, a first-hand assessment of whether colony members are variable would have been preferable. This was not done as the genus has a wide geographic distribution, with various species confined to remote locations. Nonetheless, we were able to retrieve a large amount of good-quality photos from iNaturalist, and we believe our filtering criteria allowed us to reach a valid estimate of the morphologica variability within the genus *Platycerium*. It is also important to consider that, while the genus is cultivated worldwide [48,49], not all species are easily found, and the identity of cultivars is not always known. Finally, in *P. bifurcatum* the presence of morphologically variable colony members becomes apparent in large, mature colonies [16,19], which are rare to find in cultivation, but are much easier to come across on iNaturalist.

In conclusion, this study sheds light on the evolutionary origins and life-history correlates of coloniality across the genus *Platycerium*. The evolution of colonial species with morphologically differentiated individuals was a gradual process, which started from a solitary, within-individual frond dimorphic ancestor and passed through a colonial taxon with undifferentiated individuals. Both coloniality and the presence of morphologically variable colony members were gained and lost multiple times across the phylogeny. These results combine with a growing body of literature investigating group living and division of labour in plants [16,19,39] and provides much-needed insight into the evolution of coloniality in plants.

## Supporting information

**S1 File. Supplementary 1: Supplementary tables and figures.**
(DOCX)

**S2 File. Platycerium alignment.fasta: Alignement used to build the Platycerium tree.**
(FASTA)

**S3 File. Platy_tree.nexus: Phylogenetic tree in nexus format used for the analyses.**
(NEXUS)

## Author contributions

**Conceptualization:** Riccardo Ciarle, Kevin C. Burns.

**Data curation:** Riccardo Ciarle, Katrijn de Bock.

**Formal analysis:** Riccardo Ciarle.

**Funding acquisition:** Kevin C. Burns.

**Investigation:** Riccardo Ciarle.

**Methodology:** Riccardo Ciarle, Kevin C. Burns, Katrijn de Bock.

**Project administration:** Kevin C. Burns.

**Supervision:** Kevin C. Burns.

**Validation:** Riccardo Ciarle.

**Visualization:** Riccardo Ciarle, Katrijn de Bock.

**Writing – original draft:** Riccardo Ciarle, Kevin C. Burns.

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
