## [Decision Letter · Decision Letter 0]

Dear Dr. Ciarle,

Thank you for submitting your manuscript to PLOS ONE. After careful consideration, we feel that it has merit but does not fully meet PLOS ONE’s publication criteria as it currently stands. Therefore, we invite you to submit a revised version of the manuscript that addresses the points raised during the review process.

**Congratulations! The reviewers agree that your work is almost ready for publication. Please take their comments into consideration when making your edits. I personally enjoyed reading it and look forward to moving it along to publication as soon as possible. **

We look forward to receiving your revised manuscript.

Kind regards,

T. Alexander Dececchi, Ph.D

Academic Editor

PLOS ONE

Journal Requirements:

“This study was funded by the Royal Society of New Zealand Marsden Fund (E3888, Eusociality in plants).”

**Additional Editor Comments:**

Great job. 

Reviewers' comments:

Reviewer's Responses to Questions

**Comments to the Author**

1. Is the manuscript technically sound, and do the data support the conclusions?

Reviewer #1: Yes

Reviewer #2: Yes

2. Has the statistical analysis been performed appropriately and rigorously?

Reviewer #1: I Don't Know

Reviewer #2: Yes

3. Have the authors made all data underlying the findings in their manuscript fully available?

Reviewer #1: No

Reviewer #2: Yes

4. Is the manuscript presented in an intelligible fashion and written in standard English?

Reviewer #1: Yes

Reviewer #2: Yes

Reviewer #1: The introduction could really benefit from some editing and reorganisation. Right now, there’s a strong focus on examples from animals, groups that are completely unrelated both phylogenetically and morphologically, which feels a bit off-topic and might confuse readers who are mainly interested in fern ecology. I’d suggest cutting most of these animal references and, at most, including a simple sentence like: “Although intraspecific cooperation is common in many animal groups, this phenomenon is less often described in plants.” Anything beyond that tends to shift attention away from the main subject of the study. If I start reading your abstract with examples about animals, I will think it is not a paper about plant ecology. The star of your paper is a fern genus, not animals.

It would be helpful if the introduction then moved on to clearly describe the morphology and distribution of Platycerium ferns, and then separately present the ecological context and what is already known about leaf dimorphism and variation in the group. At the moment, these elements are a bit mixed together, which makes it harder to follow. A figure or diagram showing the colony structure—maybe by adapting Figure 1 with some labels—would really help readers visualise what’s being discussed. Botanical terms like “epiphyte” don’t need to be defined, but when it comes to more specific morphological traits of this group, a little more detail would be very useful and relevant.

The methods section is good, and most definitions are clear. Just one point of clarification: are all classifications based on Hoshizaki and Price (1990)? The way it's written right after citing Hoshizaki (1990) might be a bit confusing.

I do have a small concern about the use of “polymorphic” and “dimorphic.” Since all Platycerium species are already known to be dimorphic, calling individual fronds “polymorphic” could create confusion. Maybe using “variable” for individual fronds would be a simpler alternative that avoids overlap with the established terminology. You could maybe say in the introduction that leaf dimorphism is one thing, and considering the dimorphism, every one of the two frond types may have size variations, and not "polymorphism". I do not think you used a wrong terminology, but avoiding any confusion will make the text easier to follow to people that are not familiar with these ferns.

Regarding the use of iNaturalist as a data source—I know some reviewers might raise questions, but I think it’s great that the authors used it. With proper curation and strict inclusion criteria (which they applied), it’s a valuable and underused resource for gathering real-world data. The benefits, in this case, seem to outweigh the limitations.

The phylogenetic analysis looks consistent overall, but I’d recommend including a few extra details, such as matrix size (or alignment as supporting data) or whether any trimming was done, just to make the methods more transparent.

The results are good. That said, I don’t fully agree with the statement that the phylogeny is entirely consistent with Xue et al.’s paper. There are a few differences—small, but worth pointing out. It’d be good for the authors to acknowledge these minor incongruences directly.

One specific suggestion: in this sentence “Similarly, the ancestor of all extant Platycerium species was estimated as probably colonial…” the ball plot actually suggests it’s nearly equally likely that the ancestor of clade CBF was solitary, and that P. coronarium later regained coloniality. I would recommend a higher threshold to define an ancestral state, because something just slightly above 50% is not enough. These subtle differences are important and add nuance, making the findings even more interesting and worth to be more explored in the future.

The discussion currently starts by repeating a lot of what’s already in the results or showing new results. Some of those clearer interpretations—like the number of trait gains/losses or the identification of potential synapomorphies—could actually be moved into the results section. They are an objective description of your results, not a discussion. That would also help expand the results a bit, which feel a bit too short right now.

The real discussion starts around line 271, where the authors begin exploring the evolutionary implications of coloniality. This is the part where they can really dig into what their findings mean. It might also be a better place to talk more about what “coloniality” means in a botanical context. Right now, that’s only mentioned briefly in the intro, mostly with animal comparisons, and could use more plant-focused framing.

The paragraph around lines 309–317 comes off as a negative critic about the analysis. It’s important to be honest about limitations, but ending on a negative note undersells the value of the work. I’d suggest balancing that by also pointing out the strengths of using iNaturalist data—like the richness of photos, often taken by experienced field scientists, and the careful filtering the authors did.

In general, I feel like the introduction isn’t doing justice to what the paper actually achieves. It seems aimed at attracting a zoological audience, but realistically, it’s botanists and plant ecologists who will be most interested in this study. The work itself is elegant and clever, showing how well digital photographic data can be used in botany. The suggestions above are mostly about improving the flow and structure of the text. Minor revisions are needed, just some thoughtful polishing to make the paper clearer and more focused for the right audience. I think it has strong potential for publication once those points are addressed.

Reviewer #2: This manuscript investigates trait evolution along the phylogeny using Platycerium as a testing case. I find it particularly interesting that the authors draw upon hypotheses from animal ethology to explain the morphological and functional differentiation of leaves in Platycerium. Although I am not entirely certain whether these zoology-derived hypotheses are fully applicable or compatible with plant systems,

I consider this a valuable and thought-provoking attempt.

Overall, the analyses appear sound, and the conclusions are reasonable. My primary concern lies with the structure and logic of the Introduction section, which I believe requires substantial revision. I recommend that the authors first provide a concise review of the relevant hypotheses or theoretical frameworks, followed by a more integrated introduction to Platycerium. In its current form, the division of the Platycerium background into two separate paragraphs creates confusion as to whether the study is testing a single hypothesis or several alternative ones.

One minor suggestion: I recommend that the authors provide a table—perhaps as supplementary material—listing sequence information such as GenBank accession numbers. Although these data are available online, including them would greatly facilitate access for readers who wish to trace or replicate the phylogenetic analyses.

**Do you want your identity to be public for this peer review?** For information about this choice, including consent withdrawal, please see our Privacy Policy

Reviewer #1: No

Reviewer #2: No

---

## [Author Response · Author response to Decision Letter 1]

18 Jun 2025

A point to point response to all editor's and reviewers' comments can be found in the uploaded file "response to reviewers"

---

## [Decision Letter · Decision Letter 1]

Evolutionary origins and life-history correlates of coloniality in the epiphytic fern genus Platycerium (Polypodiaceae)

PONE-D-25-23694R1

Dear Dr. Ciarle

We’re pleased to inform you that your manuscript has been judged scientifically suitable for publication and will be formally accepted for publication once it meets all outstanding technical requirements.

Kind regards,

T. Alexander Dececchi, Ph.D

Academic Editor

PLOS ONE

Additional Editor Comments (optional):

Congratulations!

Reviewers' comments:

Reviewer's Responses to Questions

**Comments to the Author**

Reviewer #1: All comments have been addressed

Reviewer #2: All comments have been addressed

2. Is the manuscript technically sound, and do the data support the conclusions?

Reviewer #1: Yes

Reviewer #2: Yes

3. Has the statistical analysis been performed appropriately and rigorously?

Reviewer #1: Yes

Reviewer #2: Yes

4. Have the authors made all data underlying the findings in their manuscript fully available?

Reviewer #1: Yes

Reviewer #2: Yes

5. Is the manuscript presented in an intelligible fashion and written in standard English?

Reviewer #1: Yes

Reviewer #2: Yes

Reviewer #1: The authors have addressed all my previous comments. I find this revised version of the manuscript to be much clearer and more objective. It is an excellent contribution to the Biology of Ferns. Therefore, I recommend that the editor accepts the manuscript for publication.

Reviewer #2: Thank the authors for your response, and I am glad to see that the two reviewers' concerns have been addressed by the authors. Interesting work in ferns.

**Do you want your identity to be public for this peer review?** For information about this choice, including consent withdrawal, please see our Privacy Policy

Reviewer #1: No

Reviewer #2: No

---

## [Editor Report · Acceptance letter]

PONE-D-25-23694R1

PLOS ONE

Dear Dr. Ciarle,

I'm pleased to inform you that your manuscript has been deemed suitable for publication in PLOS ONE. Congratulations! Your manuscript is now being handed over to our production team.

Kind regards,

on behalf of

Dr. T. Alexander Dececchi

Academic Editor

PLOS ONE